# High-Resolution Nitrogen Dioxide Measurements from an Airborne Fiber Imaging Spectrometer over Tangshan, China

**Xiaoli Zhang [1], Liang Xi [2], Haijin Zhou [2], Wei Wang [2], Zhen Chang [2], Fuqi Si [2] and Yu Wang [1],***

[1] Institutes of Physical Science and Information Technology, Anhui University, Hefei 230601, China; q20101018@stu.ahu.edu.cn
[2] Key Laboratory of Environmental Optics and Technology, Anhui Institute of Optics and Fine Mechanics, Hefei Institutes of Physical Science, Chinese Academy of Sciences, Hefei 230031, China; lxi@aiofm.ac.cn (L.X.); hjzhou@aiofm.ac.cn (H.Z.); wwang@aiofm.ac.cn (W.W.); zhchang@aiofm.ac.cn (Z.C.); sifuqi@aiofm.ac.cn (F.S.)
* Correspondence: yuwang@ahu.edu.cn; Tel.: +86-138-0125-1581

**Abstract:** The pollution caused by nitrogen dioxide is a major environmental problem in China. This study introduces a new type of atmospheric trace gas remote-sensing instrument, an airborne fiber imaging spectrometer. This spectrometer has a spectral range of 300–410 nm and works in push-broom mode with a 30° field of view on a flight path. Flight experiments were conducted on 30 December 2022 and 5 January 2023, covering heavily polluted areas east of Beijing and Tangshan. This equipment obtained the density distribution of $NO_2$ over the flight area. The results showed that pollution was mainly concentrated in the Caofeidian area and at the power station in the north, and the main source of pollution was anthropogenic. Satellite and airborne data near the pollution points were compared, and the two datasets showed a positive correlation with a correlation coefficient of 0.78 and 0.7, on the two days, respectively. This study demonstrates the capability of an airborne fiber imaging spectrometer for $NO_2$ regional emission remote sensing and identifying the pollution points.

**Keywords:** airborne; differential optical absorption spectroscopy (DOAS); imaging differential absorption spectrometer; $NO_2$

## 1. Introduction

Nitrogen oxides ($NO_x$), including NO and $NO_2$, have an impact on both the atmosphere and human health. Their main sources include industrial emissions, coal-fired combustion, and vehicle exhaust emissions. As the main representative of anthropogenic pollutant emissions, $NO_x$ emissions are significantly enhanced in urban areas. Compared with NO, $NO_2$ is more stable in the atmosphere and participates in the chemical reactions of multiple organic compounds, which is also one of the main causes of urban acid rain. Therefore, measuring $NO_2$ in the environment is a crucial aspect of air pollution monitoring.

Owing to the complex terrain and uneven population distribution in China, the atmospheric environment is complex. In the Beijing–Tianjin–Hebei region of China, the high population density and rapid economic development have contributed to compound atmospheric pollution consisting of coal smoke and motor vehicle exhaust, which display marked temporal changes. Ground-based atmospheric monitoring stations are mainly distributed in urban areas, and their detection range is limited, hindering their ability to provide large-scale data. In contrast, satellite remote sensing provides large-scale and continuous observations. However, satellite data cannot monitor real-time changes in pollutants. Therefore, airborne measurements can compensate for the shortcomings of ground-based and spaceborne observations by providing large-scale and high-temporal-resolution observations.

Previous studies in Germany, the United States, and the United Kingdom have used specially modified aircraft to conduct aerial remote sensing and the differential optical absorption spectroscopy (DOAS) technique to monitor the spatial distribution of various

pollutants in the atmosphere in real time. In 1977, Professor Platt of Heidelberg University proposed DOAS technology [1] and applied it to pollution gas tests. After many years of development, the DOAS technique was extended to mobile, shipborne, airborne, and satellite platforms to address the requirements of atmospheric observations in different regions. Furthermore, by combining DOAS with imaging technology, the two-dimensional visualization of pollution-gas distribution has been realized. For instance, on a small scale, ground-based imaging differential optical absorption spectroscopy (IDOAS) was used to study BrO formation in volcanic plumes [2–4]. Additionally, IDOAS measurements were conducted in Beijing, China, during the CARE BEIJING campaign in 2006 [5]. On a large scale, Valks et al. (2011) presented an algorithm for the retrieval of total and tropospheric $NO_2$ columns from the Global Ozone Monitoring Experiment (GOME-2) in near-real time [6], and satellite data from the SCIAMACHY were used to evaluate ship emissions based on observations of $NO_2$ distribution in the Indian Ocean or emissions from cities [7,8]. Additionally, the ability of the European Space Agency's Tropospheric Monitoring Instrument (TROPOMI) to observe the spatial and temporal patterns of $NO_2$ pollution in the continental United States was investigated by Goldberg et al. [9].

Airborne IDOAS was first applied to aircraft platforms, and the distribution of $NO_2$ density was observed near power and steel plants in South Africa [10]. The Heidelberg Airborne Imaging DOAS Instrument (HAIDI), developed by the University of Heidelberg, is capable of detecting multiple gases, such as $NO_2$, HCHO, $H_2O$, $O_3$, $O_4$, $SO_2$, and BrO [11]. The Airborne Imaging DOAS instrument for Measurements of Atmospheric Pollution (AirMAP), developed by the University of Bremen, was suitable for mapping trace gases emitted from small-scale sources with high spatial resolution. During a flight over a coal-fired power plant in northwest Germany, AirMAP detected a downwind emission plume from the exhaust stack [12]. The Atmospheric Nitrogen Dioxide Imager (ANDI), developed by the University of Leicester, discovered multiple $NO_2$ pollution points in urban and surrounding areas during a flight above Leicester [13]. The high-resolution Airborne Prism EXperiment (APEX) developed by the Royal Belgian Institute for Space Aeronomy detected $NO_2$ in polluted areas, and several flights were conducted in Belgium [14,15].

In China, the aerial remote-sensing technique for atmospheric pollutants is relatively new. Currently, suitable aerial remote-sensing equipment is rare. The Anhui Institute of Optics and Fine Mechanics conducted in-depth research on DOAS imaging technology and has achieved progress. Using airborne and ground-based platforms, a two-dimensional distribution of trace pollutant gases was obtained by Liu Jin et al. [16]. Additionally, Xi Liang et al. (2018) obtained high-resolution $NO_2$ maps using an ultraviolet–visible hyperspectral imaging spectrometer (UVHIS) in Feicheng, Shandong [17].

This study introduces a new type of imaging DOAS that differs from the previous design. The equipment adopts a combination of fiber optic and imaging spectroscopy techniques to separate the telescope from the spectrometer. The installation has lower requirements for aircraft modification, which greatly facilitates onboard installation. To test the performance of this device, it was used to detect the densities of pollutants in the area surrounding Tangshan City and obtain the density distribution of $NO_2$ over the flight path using the DOAS algorithm. Section 2 details the airborne IDOAS instrument. Section 3 describes the aircraft platform used in the experiment and the flight path. Section 4 provides the data processing procedure. Section 5 presents results and discussion, and Section 6 makes conclusions.

## 2. Instrument Details

### 2.1. Principles of Airborne IDOAS

The schematic diagram of the airborne IDOAS device is shown in Figure 1. The airborne fiber IDOAS is based on imaging differential absorption spectroscopy and fiber optics combined with the aircraft platform push scanning method. It uses an area array charge-coupled device (CCD) for rapid scanning to obtain a two-dimensional distribution of ground reflection spectra within the field of view of the flight route. The slant column

density (SCD) of the polluted gas was obtained from the spectrum using a differential absorption spectroscopy algorithm. The air mass factor (AMF) was calculated using the SCIATRAN atmospheric transport model, and the SCD was transformed into a vertical column density (VCD) that is independent of path. Finally, the flight path was obtained using the onboard Position and Orientation System (POS), and the pollution density was projected onto the map to realize the two-dimensional distribution of pollutant densities.

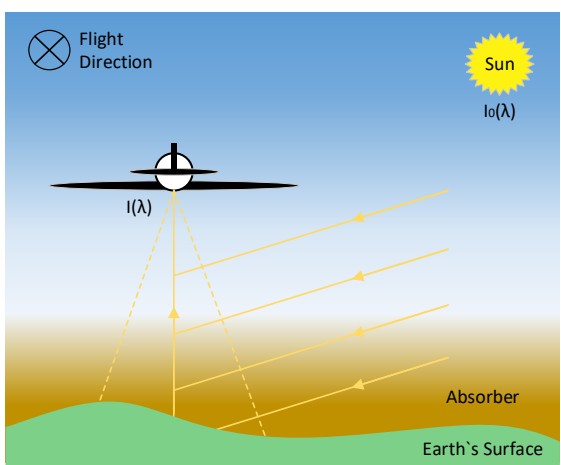

**Figure 1.** Schematic diagram of airborne imaging differential optical absorption spectroscopy (IDOAS).

### 2.2. Design of the Airborne Fiber IDOAS Instrument

As show in Figure 2, the airborne fiber IDOAS used in this experiment mainly consists of a pre-optical system, spectral imaging system, circuit control system, and power supply system. Nadir backscattered solar radiation enters the pre-optical system, and after being shaped by the pre-optical system, it is converted into electronic signals through photoelectric conversion. Parameters are shown in Table 1.

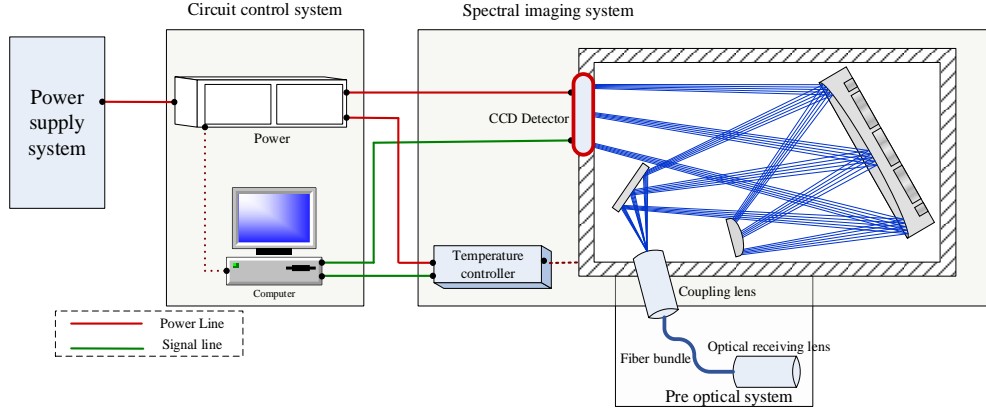

**Figure 2.** Design of airborne fiber imaging differential optical absorption spectroscopy (IDOAS) system.

**Table 1.** Parameters of the spectral imaging system.

| Parameters | Value |
| --- | --- |
| Wavelength range | 300–410 nm |
| Incidence slit | $0.05 \times 10.0$ mm$^2$ |
| Spectral resolution | 0.05 nm |
| Spatial resolution | 5 mrad |
| Detector size | $1024 \times 1024$ |
| Convex Grating Stripe Number | 2400/mm |
| Volume | $237.6 \times 142.3 \times 138$ mm$^3$ |

### 2.2.1. Pre-Optical System

To satisfy the installation requirements of airtight cabin aircraft platforms, an airworthiness modification scheme based on optical fiber transmission was proposed for pre-optical systems. The pre-optical system comprises of the following two parts: a large-field ultraviolet lens and a transmission fiber bundle. The large-field ultraviolet lens adopts a multi-lens combination structure and shapes the nadir backscattered solar radiation within the field of view. The transmission fiber consists of 50 multi-fibers, each with a 180 μm core diameter and a 10 μm cladding thickness. The fibers are spaced 200 μm apart. The width of each fiber bundle is 10 mm. Each optical fiber is an independent channel for transmitting light. The all-reflective structure of each optical fiber, composed of a core and cladding, ensures that there is no light leakage or crosstalk effect during light transmission. The UV telescope and fiber optic bundle have the same numerical aperture to minimize energy loss and prevent stray light caused by a mismatch between them.

The combination of a large-field ultraviolet lens and a fiber bundle has advantages, such as high sensitivity, strong anti-interference ability, long service life, adjustable geometric shape according to environmental requirements, and low energy loss.

### 2.2.2. Spectral Imaging System

Figure 3 shows the internal structure of the spectral imaging system. It consists of an entrance slit, a reflection mirror, a concave mirror, a convex grating, and a CCD detector. The spectrometer was designed with a Littrow–Offner structure. It also has the advantages of a Littrow-Offner-type optical system, such as a large relative aperture, small inherent aberration, and high imaging quality. The structure is compact, simple, light, and small in volume, making it suitable for airborne systems.

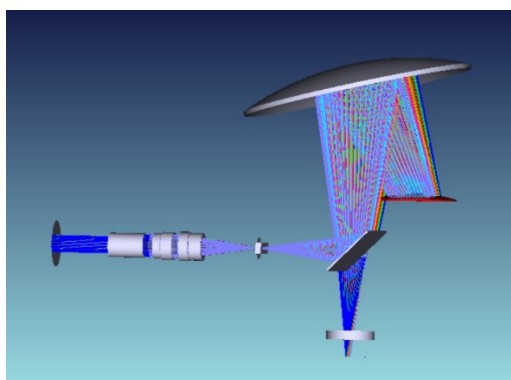

**Figure 3.** Optical simulation of the spectrometer.

Similarly, the spectrometer and the fiber optic bundle have the same numerical aperture, minimizing energy loss and preventing stray light due to any mismatch.

### 2.2.3. Circuit Control System

The circuit control system comprises of the following three parts: the temperature control system, software, and image sensor. The temperature control system is responsible for controlling the temperature of the spectrometer to reduce temperature drift. The software controls and monitors the overall equipment status. The image sensor mainly collects and stores data.

The airborne fiber IDOAS uses a CCD47-20 chip produced by E2V(UK) as an image sensor. The CCD47-20 chip adopts a backlit technique and an extremely low-noise output amplifier, making it suitable for being used in various detection fields. The CCD47-20 is a frame-transfer array CCD that mainly consists of an exposure area, a storage area, and a horizontal readout area. The size of the exposure area is 13 × 13 mm, which contains 1024 × 1024 valid pixels.

## 3. Aviation Platform and Flight Lines

### 3.1. Modification of Aircraft Platforms

The aircraft platform used in the experiment was a sealed cabin transport aircraft (AirKing350ER; NO. B-300, USA). The maximum cruising speed was 160 m/s, and the maximum endurance was 7 h. Owing to the introduction of the pre-optical fiber system, the fiber IDOAS system was installed in the aircraft in a manner that did not influence the stability of the platform (Figure 4). The main instrument was installed in the aircraft cabin, while the pre-optical fiber system was mounted on the bottom of the aircraft through a designated viewing window.

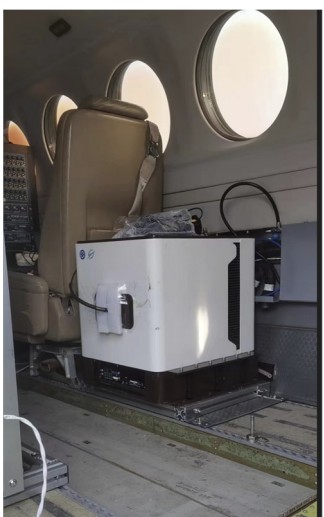

**Figure 4.** Actual installation of imaging differential optical absorption spectroscopy (IDOAS) equipment in the aircraft cabin.

### 3.2. Flight Lines

Prior to conducting flight experiments, it is necessary to plan the flight lines. The observation area was determined based on the distribution of atmospheric $NO_2$ tropospheric column densities observed by satellites in Hebei Province.

Figure 5 shows the distribution maps of the $NO_2$ tropospheric column densities in the region during November and December of 2022, derived from monthly averages of TROPOMI tropospheric $NO_2$ column concentration data [18,19]. The $NO_2$ column densities in Tangshan City were high in autumn and winter. In winter, the $NO_2$ column densities in the Caofeidian Oilfield gradually increased, and the column densities were the highest in December. Therefore, we selected the area around Tangshan City as the study region.

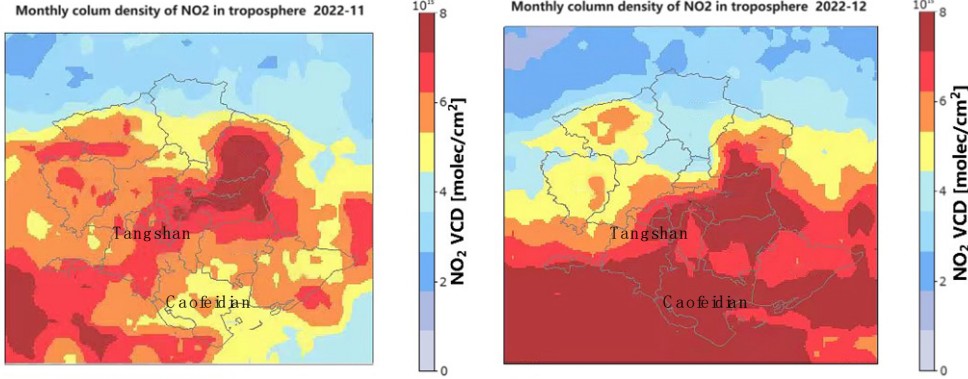

**Figure 5.** Distribution of $NO_2$ tropospheric column densities in November and December of 2022.

The experiments were better conducted under clear conditions. The experiment was conducted on 30 December 2022 and 5 January 2023, with a total of four flights. The weather on 30 December 2022 was clear and cloudless, and that on 5 January 2023 was cloudy. The experimental area covered the Caofeidian District of Tangshan City and the neighboring rural areas of Daxinzhuang Town. The instrument was mounted on the King Aircraft platform, which flew at an altitude of 700–1000 m in the scanning area at a flight speed of approximately 100 m/s. Under these parameters (altitude: 1000 m, speed: 100 m/s), the spatial resolution of the instrument was approximately 40 m in the across-track direction and 170 m along the flight direction. The scanning field of the spectrometer was 30°.

The first flight in December took off from Beijing Shahe Airport at 10:11 (local time) on 30 December 2022, and the flight duration was 2 h and 24 min (Figure 6). After taking off, the aircraft flew to Tangshan at an altitude of approximately 2260 m. Once reaching Tangshan, the aim of this flight was to scan the area with the southern oil fields and northern steel plants, and an altitude of approximately 700–1000 m and a speed of approximately 100 m/s were maintained. The second flight took off from Beijing Shahe Airport at 15:33 on 30 December 2022, and the flight duration was 2 h and 9 min (Figure 7). After taking off, the aircraft flew to Tangshan at an altitude of approximately 2280 m. Once reaching Tangshan, the aim of this flight was to scan the northern steel plants in Tangshan City. The altitude of the flight was approximately 700–1000 m, and it maintained a speed of approximately 100 m/s. The experimental area had flat terrain with an elevation of approximately 0–30 m.

The first flight in January took off from Beijing Shahe Airport at 10:31 on 5 January 2023, and the flight duration was 3 h and 7 min (Figure 8). After taking off, the aircraft flew to Tangshan at an altitude of approximately 2300 m. Once reaching Tangshan, the altitude of the flight was maintained at approximately 700–1000 m, with a speed of approximately 100 m/s. The second flight took off from Beijing Shahe Airport at 15:55 on 5 January 2023, and the flight duration was 1 h and 46 min (Figure 9). After taking off, the aircraft flew to Tangshan at an altitude of approximately 2300 m. Once reaching Tangshan, the altitude of the flight was maintained at approximately 700–1000 m, with a speed of approximately 100 m/s. The aim of the flights was to scan the area with the chemical and steel plants in Tangshan City at different altitudes. The experimental area had flat terrain, with an elevation of approximately 0–30 m.

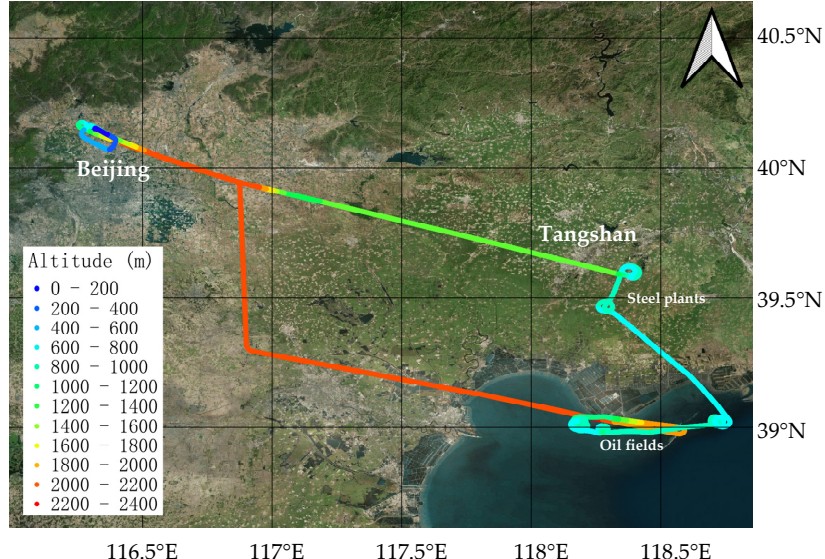

**Figure 6.** The flight path on the morning of 30 December 2022.

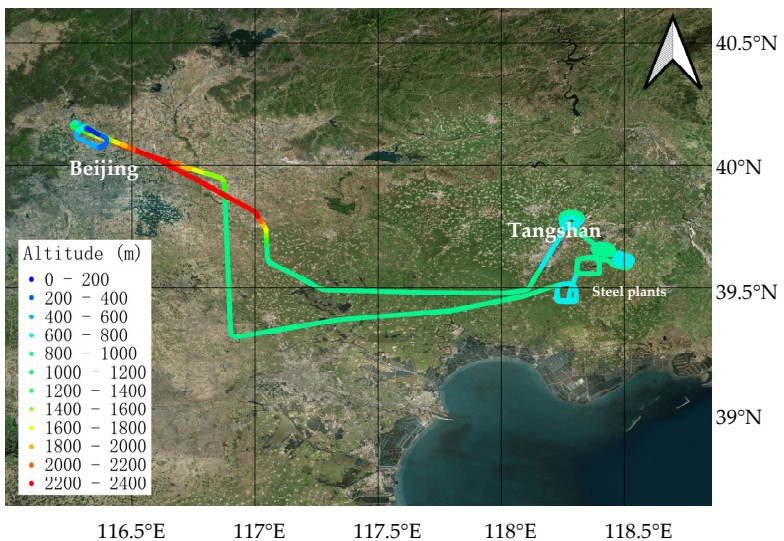

**Figure 7.** The flight path on the afternoon of 30 December 2022.

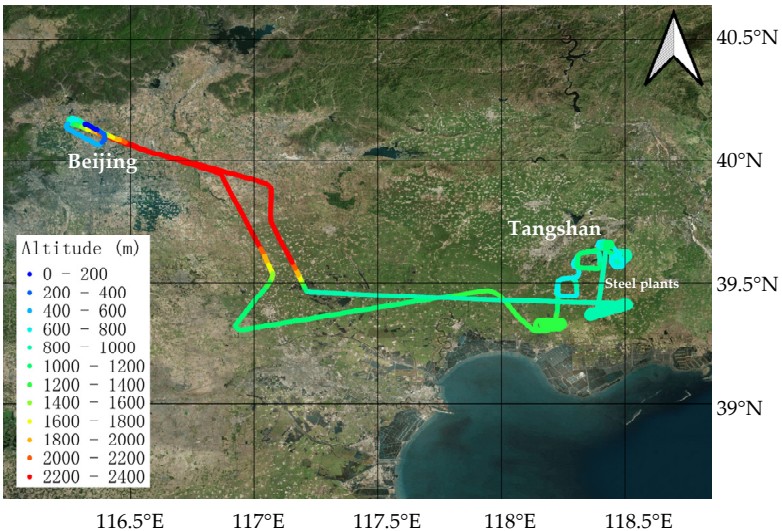

**Figure 8.** Flight path on the morning of 5 January 2023.

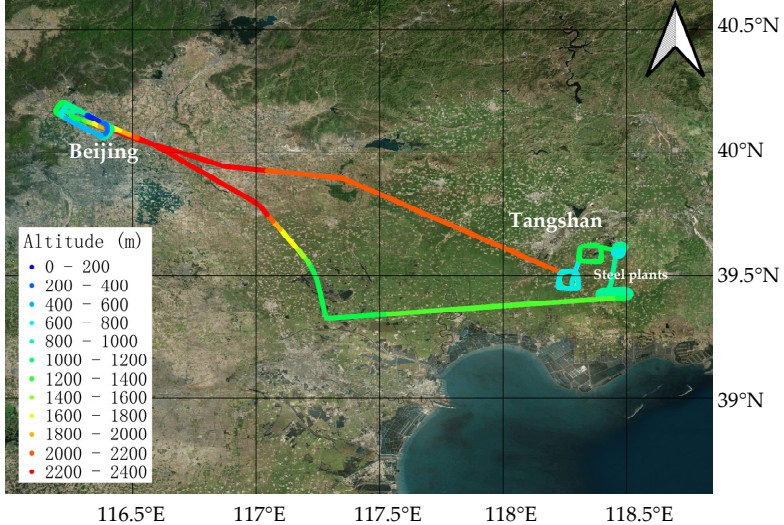

**Figure 9.** Flight path on the afternoon of 5 January 2023.

## 4. Data Processing

The algorithm for the retrieval of the VCD of $NO_2$ in the troposphere for airborne fiber IDOAS includes four steps, as shown in Figure 10. The first step involves the necessary data preprocessing procedures in order to make spectral data from the raw electrical data collected by the detector. Subsequently, after preprocessing, in the second step, an established DOAS technique was used to analyze the airborne fiber IDOAS spectral data in an appropriate wavelength region to obtain the SCDs of the target gases. In the third step, the AMFs were calculated, and the SCDs were converted to VCDs for each observation using the SCIATRAN radiative transfer model. Lastly, in the fourth step, the $NO_2$ VCDs were geo-referenced and overlaid onto Google satellite map layers by using the POS data from the sensors.

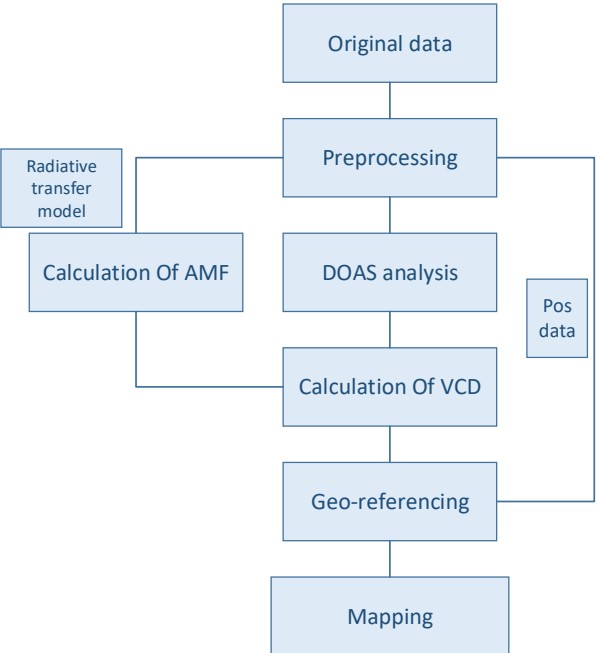

**Figure 10.** Data processing.

### 4.1. Preprocessing

Before performing spectral analysis, the data were preprocessed. This included data selection, dark current correction, spatial binning, and in-flight calibration.

#### 4.1.1. Data Selection

Combined with the onboard POS data, invalid data from before takeoff and after landing were excluded. Additionally, spectral data collected during takeoff when the aircraft did not reach the predetermined height of 2000 m and the pitch angle was greater than $4°$, and during landing when the altitude was lower than 2000 m and the pitch angle was lower than $-2°$, were excluded, owing to the large changes in the altitude of the aircraft.

#### 4.1.2. Dark Current Correction

To reduce the effect of the detector's dark current on the signal, dark current correction is required. We performed a dark current correction by blocking the fore-optics based on the measurement taken at the beginning of the entire flight to improve instrument performance and reduce the errors in the DOAS fit. Figure 11 shows the spectral intensity comparison before and after dark current correction.

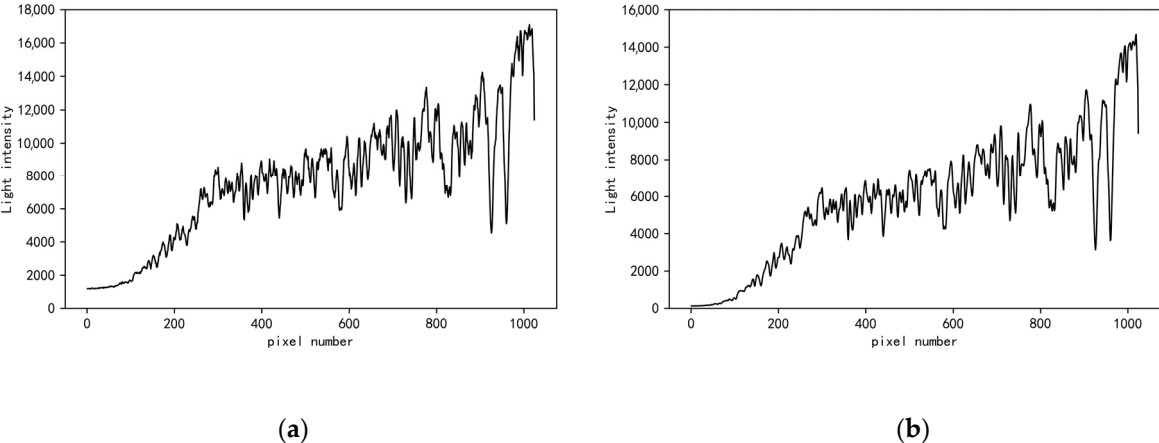

(**a**)

(**b**)

**Figure 11.** Spectral graph before and after the dark current correction. (**a**) Original spectra; (**b**) Spectra after dark current correction.

The CCD is cooled using TE cooling to reduce dark background noise. Figure 11a shows a single-line spectrum of the detector before preprocessing, which is relatively noisy. Binning is performed to effectively suppress the noise and improve the spectral quality. The spectral noise level varies with the solar azimuth angle and ground conditions during the flight. The signal-to-noise ratio (SNR) level was approximately 900:1 under the conditions of a 60° solar zenith angle and 0.3 surface albedo.

### 4.1.3. Spatial Binning

Figure 12 shows the solar scattering spectral intensity collected by the detector during the aircraft flight. Owing to the use of an optical fiber, the image detected by the detector shows a strip distribution. The bright and dark stripes are initially segregated. The bright stripes correspond to 13 rows of pixels, while the dark stripes correspond to 2 rows of pixels. The main energy within a single fiber is contained in the bright stripes, while the dark stripes are the overlapped information between adjacent fibers. The bright stripes are spatially integrated into pixel elements, whereas the dark stripes are effectively eliminated during data processing. Consequently, each fiber represents a spectrum with a high signal-to-noise ratio within its field of view.

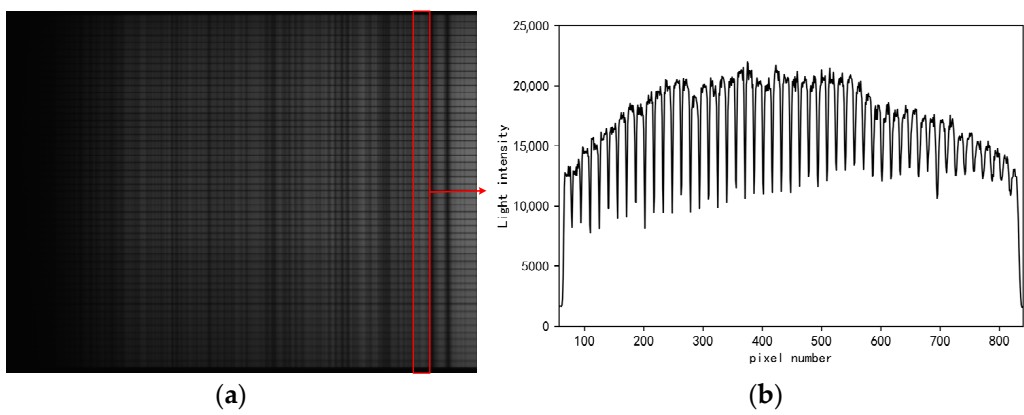

(**a**)

(**b**)

**Figure 12.** Intensity map of array charge-coupled device (CCD). In (**a**), the vertical dimension of the spectrum is the spatial dimension, and the horizontal dimension is the spectral dimension; (**b**) Distribution of spectral intensity along the vertical spatial dimension.

To increase an instrument's signal-to-noise ratio (SNR) and sensitivity to $NO_2$, raw DOAS imaging pixels are typically aggregated in the across-track direction. In this study, four sets of fiber optic spectral data were binned in an across-track orientation during the

data analysis to increase the SNR, and the single group field of view angle was 2.4°, with a spatial resolution in the across-track direction of approximately 40 m when the altitude was approximately 1000 m.

### 4.1.4. In-Flight Calibration

In-flight wavelength calibration is crucial for subsequent DOAS analysis because the wavelength-to-pixel registration and slit function shape of the airborne fiber IDOAS may differ from the laboratory calibration results. To obtain this in-flight wavelength calibration, the observed spectra were fitted to a high-resolution solar reference spectrum using a slit-function correction and wavelength shift [20]. The laboratory calibration determined the nominal wavelength-to-pixel registration, which served as the initial value in the iterative fitting procedure for converging to the optimal solution. The spectrum was divided into n tiny intervals for translation, expansion, and compression in order to perform this in-flight wavelength calibration, as indicated by the following equation:

$$\Delta\lambda = a + b(\lambda - \lambda_0) + c(\lambda - \lambda_0)^2 \qquad (1)$$

where $\lambda$ represents the correction wavelength, $\lambda_0$ represents the center wavelength of the n_th small intervals, a represents the translation of the fitting, and b and c represent the expansion and contraction of the quadratic fitting, respectively.

The laboratory temperature was maintained at approximately 20 °C, whereas the in-flight temperature was approximately 0 °C, which is significantly different from the laboratory temperature. Consequently, the wavelength-to-pixel registration and slit function shape may alter during the experiment. Figure 13 shows the effective shifts and spectral resolutions (full-width at half maximums, FWHMs) of the various across-track positions. Figure 13a shows the offset values of the FWHMs at 338, 354, and 370 nm in different across-track directions, with an offset range of 0.38–0.49 nm. Figure 13b shows the offset values of the spectrum in across-track directions, with an offset range of −0.08–0.4 nm.

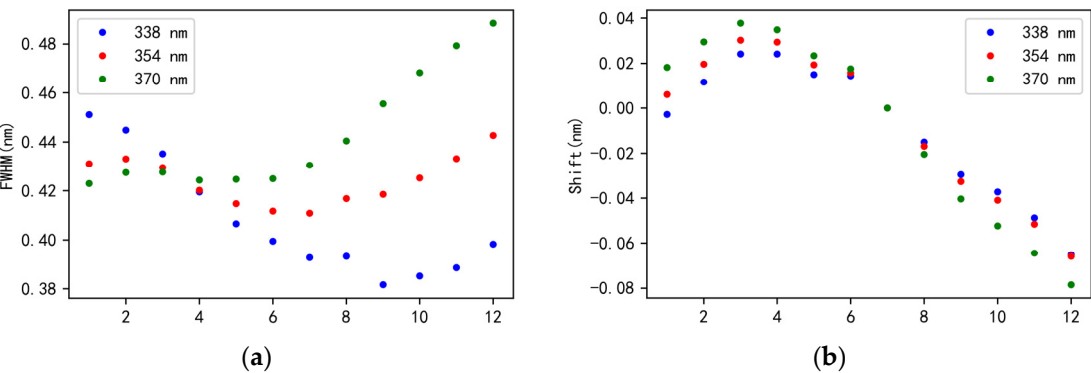

(a)          (b)

**Figure 13.** In-flight spectral calibration: (**a**) Spectral resolution, full-width at half maximums (FWHMs); (**b**) Spectral shift at various across-track positions.

### 4.2. DOAS Analysis

To retrieve the $NO_2$ SCD, the observed spectra of the airborne fiber IDOAS were analyzed using the QDOAS 3.2 software [21]. Table 2 lists the details of the DOAS analysis settings. Considering the strong $NO_2$ absorption features, the fitting window was within the 338 and 370 nm wavelength regions. For each spectrum, the direct output of the DOAS fit was the differential SCD, which is the difference between the $NO_2$ integrated density along the effective light path of the studied spectrum and the selected reference spectrum. Reference spectra are typically obtained over a clean rural area. For example, in the experiment on the morning of 30 December 2022, the spectra collected in the north of the flight path were chosen as the reference spectra. QDOAS 3.2 software also provides the RMS of the residuals and the retrieval error.

**Table 2.** Differential optical absorption spectroscopy fitting parameters.

| Parameter | Data Source |
|---|---|
| Wavelength | 338–370 nm |
| Wavelength calibration | Kurucz |
| Absorption cross-section | |
| $NO_2$ | Vandaele (1998): 294 k |
| $O_4$ | Hermans (2011): 298 k |
| $O_3$ | Bogumil (2003): 293 k |
| Polynomial degree | Order 5 |

Figure 14 shows a typical $NO_2$ DOAS fit and the corresponding residual spectra. The collection time was 11:26:13 and the location was 118.564°N, 39.996°E. Four sets of fiber optic spectral data were binned in the across-track direction. The differential SCD was $5.24 \times 10^{16}$ molec/cm$^2$, the RMS of the residuals was $2.17 \times 10^{-3}$, and the fit error was $4.68 \times 10^{15}$ molec/cm$^2$.

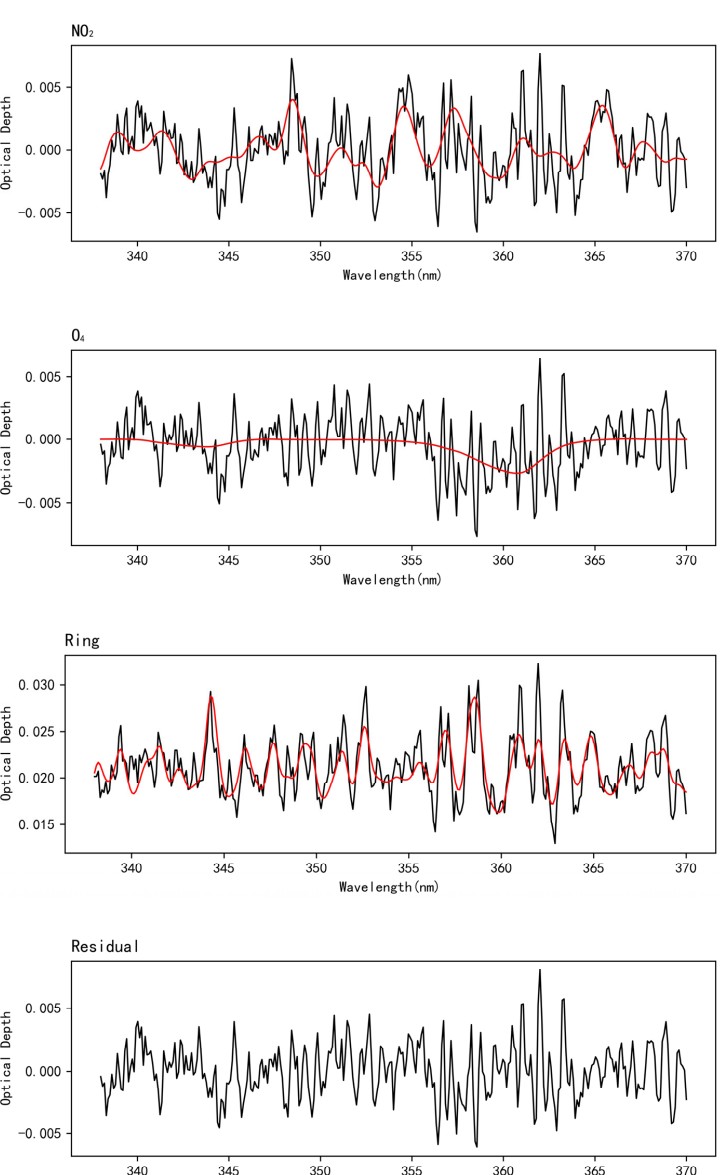

**Figure 14.** Typical sample of differential optical absorption spectroscopy (DOAS) fit for $NO_2$. The red line shows the fitted absorption structure, while the black line shows the fit result plus the residual.

### 4.3. AMF Calculations, Geo-Referencing, and Mapping

The SCD represents the integrated density along the effective light path of observation. It is strongly dependent on the viewing geometry and radiative transfer. Therefore, before drawing the projection map, it is necessary to convert the SCD into the VCD, which is path-independent. The method for converting the SCD to the VCD is based on the AMF.

The AMF is defined as the ratio of the SCD to the VCD:

$$\text{AMF} = \frac{\text{SCD}}{\text{VCD}} \tag{2}$$

Meanwhile, the AMF is influenced by several factors, including the path of light (e.g., the sun and viewing angle), trace gas and aerosol vertical profiles, and surface reflectivity. The accuracy of the AMF calculation affects the retrieval accuracy of the trace gas VCD. To enhance the accuracy of the trace gas VCD retrieval, it is necessary to consider the impact of various factors on the AMF. The AMF calculation method uses the SCIATRAN [22] radiative transfer model and establishes a lookup table (LUT). Combined with the AMF, the SCD is converted into a path-independent VCD as follows:

$$\text{VCD} = \frac{\text{SCD}}{\text{AMF}} \tag{3}$$

Figure 15 shows the calculation process of the AMF and VCD.

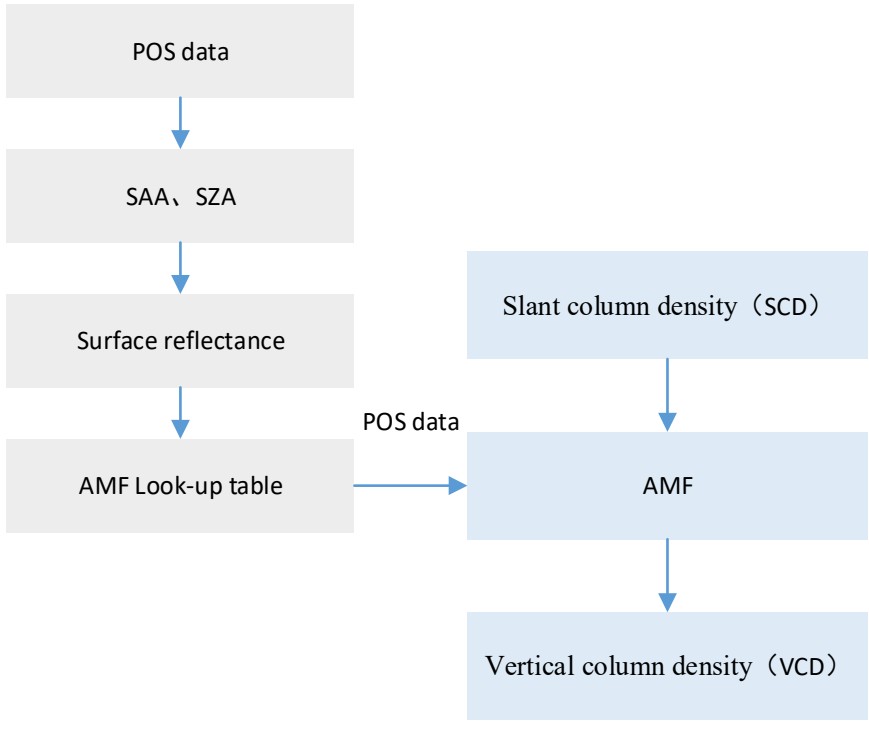

**Figure 15.** Calculation flowchart for air mass factor (AMF) and vertical column density (VCD).

In this experiment, the observation angle of the instrument was calculated based on the aircraft's POS data. The solar azimuth angle and solar zenith angle can be deduced from the latitude and longitude of each observation. For the AMF calculation, the Landsat 8 Operational Land Imager surface reflectance product was utilized, which has a wavelength range of 433–450 nm. The Aerosol Optical Depth (AOD) information for the AMF calculation was obtained from the MODIS AOD product at 412 nm and interpolated in two dimensions to each airborne ground pixel. Owing to the unavailability of the planetary boundary layer (PBL) height during the flight, a typical height of 2 km was used as a

reasonable estimate for mid-latitude areas in China. The single-scattering albedo was set to 0.93, and the asymmetry factor was set to 0.68 for the aerosol extinction profile.

Figure 16 shows the surface reflectance from 24 December 2022 to 3 January 2023, considering the altitude of the flight of the aircraft from 10:41 to 10:47. Figure 17 shows the calculated AMF for a flight line. It is clear that the AMFs are highly dependent on surface reflectance. Table 3 shows the AMF LUT parameter settings.

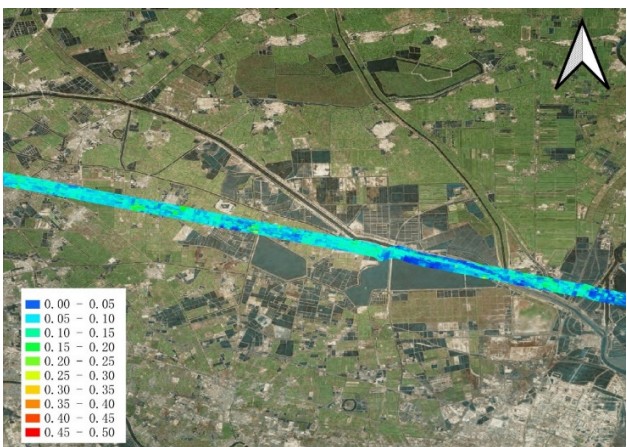

**Figure 16.** Surface reflectance from 24 December 2022 to 3 January 2023.

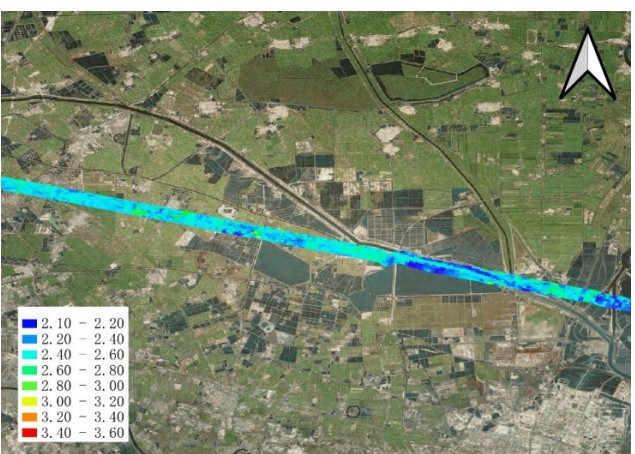

**Figure 17.** Air mass factor (AMF) on 30 December 2022.

**Table 3.** Parameters of the air mass factor lookup table (LUT).

| Parameter | Setting |
|---|---|
| Wavelength | 354 nm |
| Detector height | 1 km |
| Surface reflectance | 0.01–0.4 (steps of 0.01) |
| Solar zenith angle | 60–70° (steps of 10°) |
| Viewing zenith angle | 0–50° (steps of 10°) |
| Relative azimuth angle | 0–180° (steps of 30°) |
| Aerosol optical depth | 0–1.6° (steps of 0.1°) |
| Aerosol profile | Box of 2.0 km |
| $NO_2$ profile | Box of 2.0 km |

### 4.4. Aircraft Angle and Geolocation Correction

Owing to the instability of the aircraft during flight, there was a deviation between the ground pixels and aircraft positions. Therefore, the accurate matching of ground pixels is essential.

The altitude of an aircraft is primarily determined by its pitch, roll, and yaw angles. Conventionally, the pitch angle is considered positive when the aircraft's nose is pointing upwards, while the roll angle is positive when the right wing is pointing downward. The yaw angle is measured in degrees clockwise from north ($0°$).

Information on aircraft positioning was read before the collection of each spectrum. Therefore, the middle value between the two data collections was selected as the pixel center.

The spatial displacement of the ground in the flight direction and the spatial displacement vertically can be obtained from the pitch and roll angles as follows:

$$L = H \cdot \tan\theta, \tag{4}$$

$$d = \frac{H}{\cos\theta} \tan(\theta_i - \varnothing), \tag{5}$$

where $\theta$ represents the pitch angle, $\varnothing$ represents the roll angle, L is the ground spatial displacement in the flight direction, d is the vertical spatial displacement, H is the current flight altitude, and $\theta_i$ is the angle between the center of the field and the vertical direction.

Assuming that, at a certain point in time, the aircraft is located at a longitude and latitude $(X_0, Y_0)$ and the ground coordinates of the actual testing location are $(X, Y)$ (the displacement deviation between $X'$ and $Y'$), the relationship below can be derived:

$$X = X_0 + X', \tag{6}$$

$$Y = Y_0 + Y' \tag{7}$$

Furthermore, $(X', Y')$ can be represented as follows:

$$X' = \frac{180°}{\pi R_0 \cos Y_0}(\cos\varphi \cdot d + \sin\varphi \cdot L), \tag{8}$$

$$Y' = \frac{180°}{\pi R_0}(-\sin\varphi \cdot d + \cos\varphi \cdot L) \tag{9}$$

Additionally, based on the above equation, it can be derived that:

$$X' = \frac{180° \cdot H}{\pi R_0 \cos Y_0}\left(\cos\varphi \cdot \frac{\tan(\theta_i - \varnothing)}{\cos\theta} + \sin\varphi \cdot \tan\theta\right), \tag{10}$$

$$Y' = \frac{180° \cdot H}{\pi R_0}\left(-\sin\varphi \cdot \frac{\tan(\theta_i - \varnothing)}{\cos\theta} + \cos\varphi \cdot \tan\theta\right) \tag{11}$$

## 5. Results and Discussion

### 5.1. IDOAS Results

The $NO_2$ VCD two-dimensional distribution map obtained using the airborne fiber IDOAS for the flight on 30 December 2022 is shown in Figure 18. The VCD of $NO_2$ in the Caofeidian area was significantly high, especially over pollution points 3, 4, and 5. The highest VCD of $NO_2$ was at point 4 in the southern part of Caofeidian, where oil field drilling platforms and coal utilization companies are located. Large-scale dispersed pollution was observed in the coastal area of pollution point 3, which has artificial islands in oil fields and oil and gas transportation companies. A large area of high $NO_2$ VCD distribution was observed around pollution point 5. Considering the wind direction on this day (west and northwest wind), $NO_2$ pollution may be caused by the diffusion of pollutants from oil fields and maritime vessels. In other areas, especially Tangshan City and the surrounding areas of Beijing, the VCD of $NO_2$ was relatively low. The maximum VCD of $NO_2$ during the flight was $3.3 \times 10^{16}$ molec/cm$^2$, the minimum value was $7.36 \times 10^{15}$ molec/cm$^2$, and the average value was $1.92 \times 10^{16}$ molec/cm$^2$. Moreover, the average uncertainty of $NO_2$ during the flight was found to be $2.80 \times 10^{15}$ molec/cm$^2$. The error estimation method is adopted from reference [17].

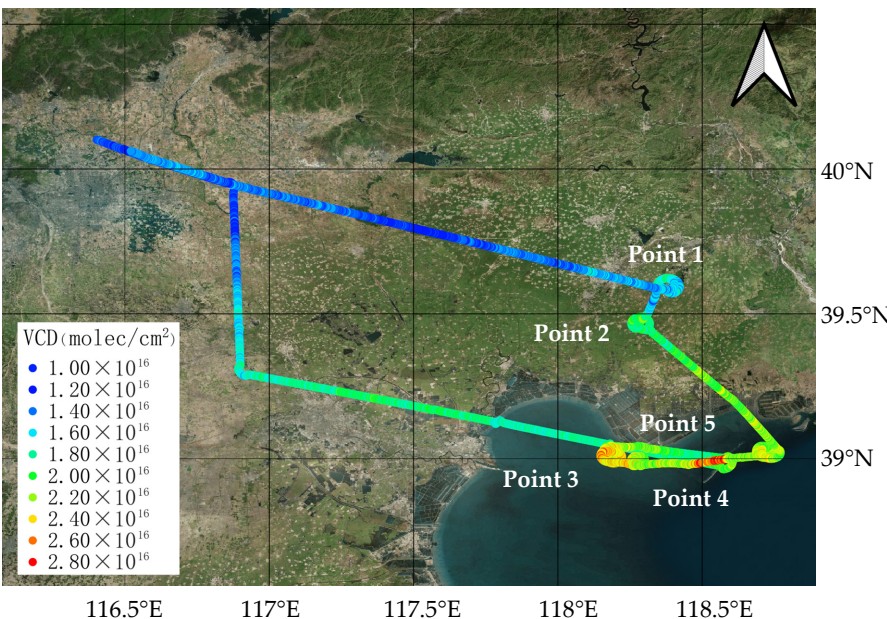

**Figure 18.** Two-dimensional distribution map of NO$_2$ vertical column density (VCD) on the morning of 30 December 2022.

The results for the afternoon of 30 December 2022 showed that the VCD of NO$_2$ was significantly high on the flight line (Figure 19). However, considering that the aircraft took off late and the light was poor, only the data collected before 17:00 were valid. A high VCD was observed at points 1 and 2. According to the analysis of the wind direction and satellite data on that day, this may be caused by the emission source of NO$_2$ in the south.

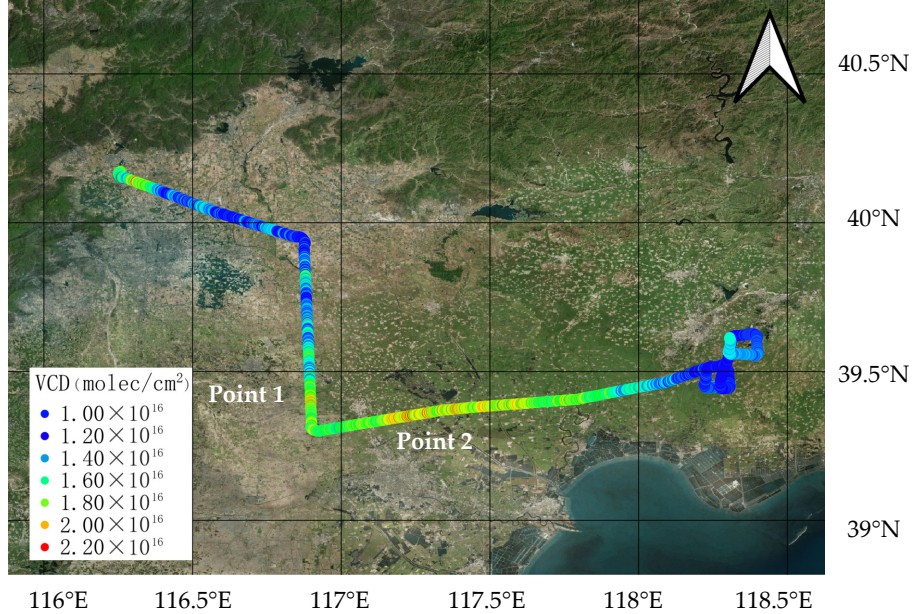

**Figure 19.** Two-dimensional distribution map of NO$_2$ vertical column density (VCD) on the afternoon of 30 December 2022.

Figure 20 shows the NO$_2$ VCD two-dimensional distribution map obtained using the airborne fiber IDOAS for the flight on 5 January 2023. On this day, the VCD of NO$_2$ remained relatively high throughout the flight. In particular, at point 1 in the north, where there were several steel plants, coking plants, and chemical plants, the VCD of NO$_2$ was relatively high. In addition, there were steel and chemical plants at points 2, 3, 4, 5, 6,

and 8. Point 7 was located on a farm and had no evident emission sources. According to the wind direction of the day (west and south winds), NO$_2$ pollution may be caused by the diffusion of pollutants from the west and the oil field drilling platform in the south. The maximum VCD of NO$_2$ during the flight was $7.16 \times 10^{16}$ molec/cm$^2$, the minimum value was $3.82 \times 10^{15}$ molec/cm$^2$, and the average value was $4.86 \times 10^{16}$ molec/cm$^2$. The average uncertainty of NO$_2$ during the flight was $3.60 \times 10^{15}$ molec/cm$^2$.

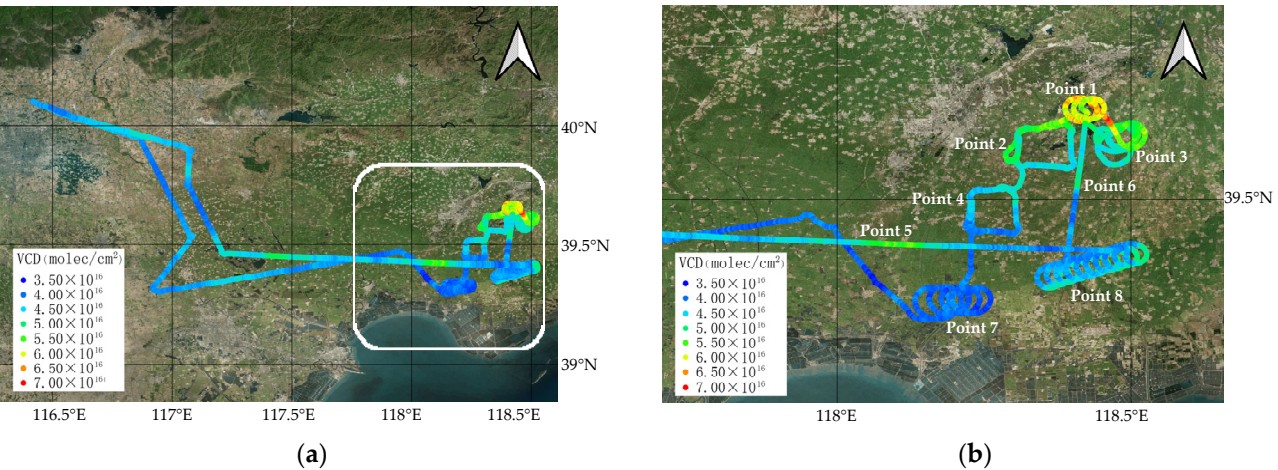

**Figure 20.** Two-dimensional distribution map of NO$_2$ vertical column density (VCD) on the morning of 5 January 2023. (**a**) VCD distribution of NO$_2$ along the flight path; (**b**) Enlarged view near the pollution points.

The results for the afternoon of 5 January 2023 showed high levels of NO$_2$ VCD in the flight path (Figure 21). However, because the aircraft took off late and the light was poor, only the data collected before 17:00 were valid. A high VCD was observed at points 1 and 2. According to the wind direction and satellite data on that day, this may be caused by the emission source of NO$_2$ in the south.

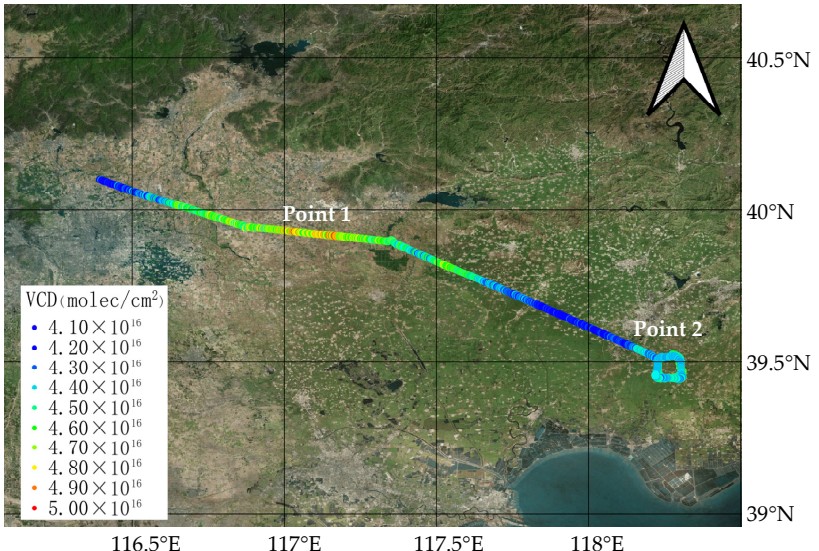

**Figure 21.** Two-dimensional distribution map of NO$_2$ vertical column density (VCD) on the afternoon of 5 January 2023.

## 5.2. Comparison of IDOAS and Satellite Data

Relative to satellite remote sensing, airborne remote-sensing technology is more sensitive to the atmospheric composition near the surface and has a higher temporal and spatial

resolution. In this experiment, tests were also performed at different altitudes around the pollution points. For instance, during the initial flight experiment on 30 December 2022, two flights were conducted at different altitudes over pollution point 3. The average VCD of $NO_2$ was $2.73 \times 10^{16}$ molec/cm$^2$ at an altitude of 900 m and $2.83 \times 10^{16}$ molec/cm$^2$ at an altitude of 740 m. There are few discrepancies between them. It is evident that $NO_2$ pollutants are primarily concentrated near the surface. Therefore, airborne observational data are suitable for verifying satellite data. In order to assess the agreement of satellite and airborne data, the airborne fiber IDOAS $NO_2$ VCDs were compared with TROPOMI data in the vicinity of the day's flight.

The results of the airborne fiber IDOAS at the airborne flight line on the morning of 30 December 2022 were compared with the TROPOMI $NO_2$ data of that day (Figure 22). The two $NO_2$ VCD datasets at the flight line show a high coincidence, with a correlation coefficient of 0.78. The $NO_2$ VCDs measured by the TROPOMI satellite at most pollution points on the flight path were slightly higher than the those of the airborne data, whereas the airborne data were slightly higher in areas with more severe pollution, such as points 4 and 5.

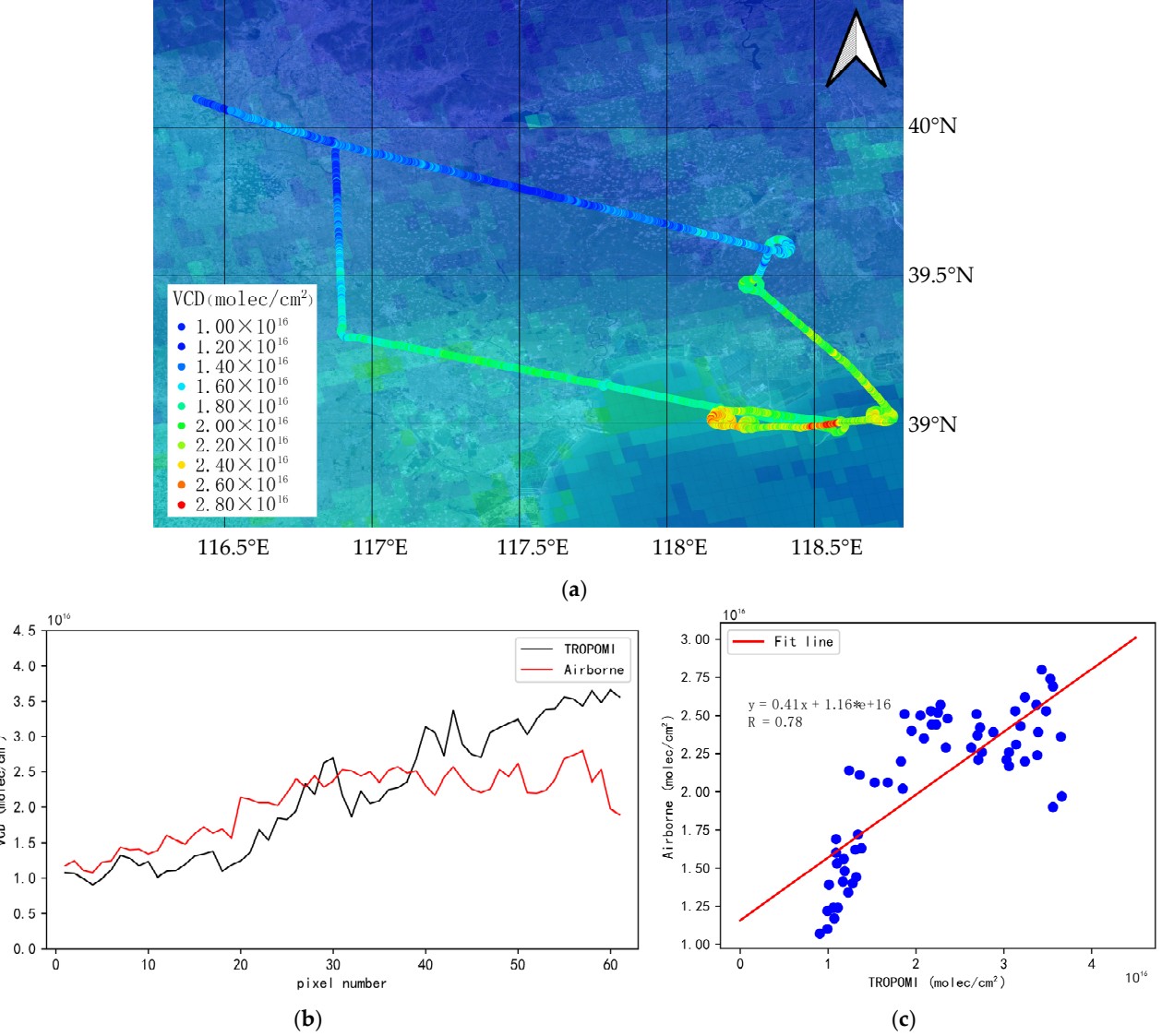

**Figure 22.** (**a**) Distribution map of airborne and satellite $NO_2$ vertical column density (VCD) on 30 December 2022; (**b**) Comparison of $NO_2$ VCDs between airborne and satellite datasets; (**c**) Analysis of the correlation between airborne and satellite data.

As shown in Figure 23, the results of the airborne fiber IDOAS at the airborne flight line on the morning of 5 January 2023 were compared with the TROPOMI data of that day. It can be seen that the two NO$_2$ VCD datasets in the flight line have high coincidence, with the correlation coefficient of 0.7. The NO$_2$ VCDs measured by the TROPOMI satellite at most pollution points on the flight line were slightly higher than those of the airborne results, whereas the airborne data were slightly higher in areas with more severe pollution, such as point 1.

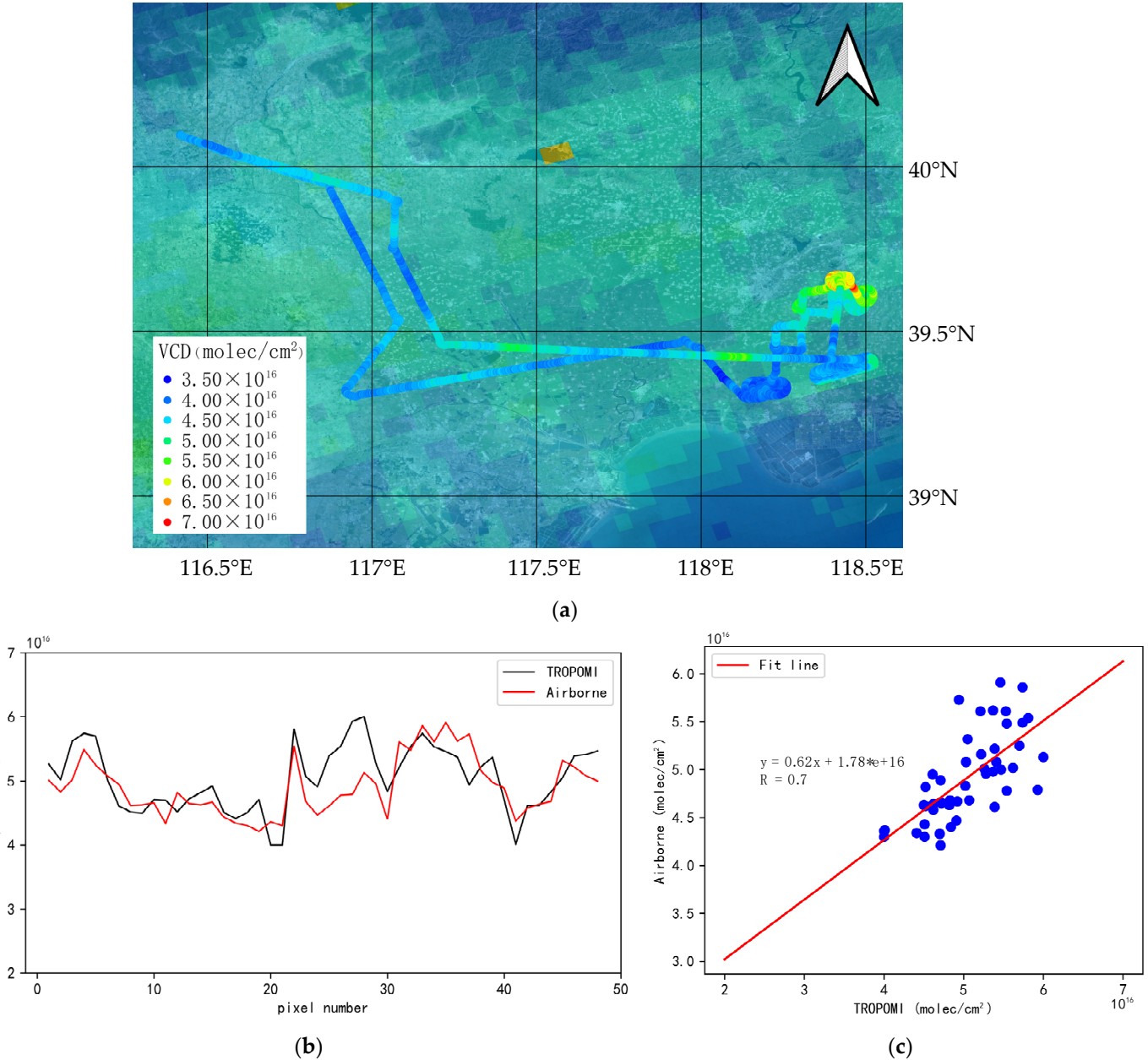

**Figure 23.** (**a**) Distribution map of airborne and satellite NO$_2$ vertical column density (VCD) on 5 January 2023; (**b**) Comparison of NO$_2$ VCDs between airborne and satellite datasets; (**c**) Analysis of the correlation between airborne and satellite data.

The distribution maps of airborne and satellite NO2 VCDs on 30 December 2022 and 5 January 2023 are shown in Figures 22a and 23a. In terms of spatial dimensions, the resolution of satellite observation is 3.5 × 5.5 km$^2$, while the airborne pixel is about 40 m × 170 m$^2$. The ground pixel of the satellite is much larger than the airborne pixel. Therefore, the airborne pixels need to be averaged within each satellite pixel for comparison.

The large discrepancy in the spatial dimensions may lead to discrepancies in the two datasets. In addition, the TROPOMI product uses the a priori TM5-MP chemical transport model to separate the stratospheric and tropospheric layers in the retrieval, while, in this study, the SCD-VCD transformation for airborne data is performed using the SCIATRAN radiative transfer model. Finally, due to the rapid changes in the distributions of gases, the difference in the observation time of the aircraft and the overpass time of the satellite also caused discrepancies in the two datasets. The three factors result in differences between the airborne data and the TROPOMI data. We can reduce the difference in the two datasets by making the aircraft observation time closer to the overpass time of the satellite, and trying to increase the flight altitude of the aircraft.

## 6. Conclusions

This study presents a newly developed airborne fiber IDOAS with a broad spectral range of 300–410 nm and a high spatial resolution. The airborne fiber IDOAS comprises a fiber transmission system and an IDOAS system, which provides advantages such as high spectral imaging resolution, a large field of view, and a compact structure.

In this experiment, the DOAS technique was used to retrieve the $NO_2$ SCD in the flight area. The SCIATRAN radiation model was then used to calculate the AMF and convert the SCD into a path-independent VCD. Finally, a map-projection algorithm was used to project the results onto a map and realize a two-dimensional distribution of the $NO_2$ VCDs.

In this study, IDOAS airborne observations were performed over Tangshan, China, on 30 December 2022 and 5 January 2023. The results of the two morning experiments were relatively ideal, whereas the afternoon flight experiment had significant errors due to poor light. On the morning of 30 December 2022, the maximum VCD of $NO_2$ during the flight was $3.3 \times 10^{16}$ molec/cm$^2$, and on the morning of 5 January 2023, the maximum VCD of $NO_2$ during the flight was $5.56 \times 10^{16}$ molec/cm$^2$.

Finally, we compared the $NO_2$ VCD dataset of the airborne fiber IDOAS with that of the TROPOMI satellite. The distribution of the $NO_2$ VCDs between the two datasets was strongly positively correlated and showed a high correlation. The correlation coefficients were 0.78 and 0.7, respectively.

Compared to the data from the satellite instruments, those from the airborne fiber IDOAS had a higher spatial resolution. Additionally, compared with ground DOAS instruments, the airborne fiber IDOAS has higher flexibility. The design of the airborne fiber IDOAS in this study compensates for the shortcomings of the ground equipment and onboard instruments. This study demonstrates the ability of the airborne fiber IDOAS to locate $NO_2$ pollution points.

**Author Contributions:** Conceptualization, H.Z. and F.S.; methodology, X.Z. and H.Z.; software, H.Z., Z.C. and L.X.; validation, X.Z. and L.X.; formal analysis, X.Z., H.Z. and L.X.; investigation, X.Z. and W.W.; resources, L.X. and Y.W.; data curation, H.Z., Z.C. and F.S.; writing—original draft preparation, X.Z.; writing—review and editing, X.Z., W.W. and H.Z.; visualization, X.Z.; supervision, F.S. and Y.W.; project administration, F.S.; funding acquisition, F.S. All authors have read and agreed to the published version of the manuscript.

**Funding:** This research was funded by the National Key Research and Development Program of China (Grant Number 2023YFC3705104).

**Data Availability Statement:** The data presented in this study are available upon request from the corresponding author. The data are not publicly available due to very big data.

**Acknowledgments:** We are grateful to the BIRA for providing the QDOAS software.

**Conflicts of Interest:** The authors declare no conflicts of interest.

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
