# Peer review of "High-Resolution Nitrogen Dioxide Measurements from an Airborne Fiber Imaging Spectrometer over Tangshan, China"

_remotesensing, doi:10.3390/rs16061042_

Round 1

Reviewer 1 Report

Comments and Suggestions for Authors

Review of “High-resolution nitrogen dioxide measurements from airborne fiber imaging spectrometer over Tangshan, China” by Xiaoli Zhang et al.

The article describes a new airborne instrumentation to measure NO2 in the atmosphere by measuring Nadir backscattered solar radiation by mounting onto an airplane flying about 1 Km altitude. The measurements have high spectral (~0.4 nm) as well as spatial resolution (40 m x 170 m). The description of the instrumentation, measurement and retrieval methods were described. Two measurement data were compared with the TROPOMI Satellite data which compared well. The article is well written, and I suggest accepting it for publication with minor edits. My comments and queries are listed below:

1.     The IDOAS instrument described here is very similar to the HAIDI (General et al., 2014 quoted in the Reference section) based on Fig. 2 and 3.

a.     What is the difference between the instrumentation in terms of design, measurements and retrieval?

b.     Compared to HAIDI what is innovative in the new design?

c.     Like in HAIDI the authors used 330 – 370 nm band for NO2 measurements. The peak of absorption is at 405 nm, and that region will have minimal spectral interference from other common air pollutants. Why is this peak of the NO2 absorption spectrum not used for improving sensitivity?

2.     Can you give a reference from which Figure 5 was adopted?

3.     Figure 11 shows a basic spectral measurement data. This shows a noisy spectrum. Is the CCD array TE cooled? What is the noise level? How is it affecting the measurement sensitivity (of NO2 concentration)? In fact, noise characterisation and error estimation method (accuracy as well as precision) must be described.

4.     Since the measurement range is extended beyond red region, one can use the oxygen lines (say B-band) to normalise the spectral measurements as well as for any corrections to the retrieved concentrations as an automatic check.

5.     On page 10 line 304 – 305, an example retrieval result is mentioned. It is better to give an error estimate in the same units of concentration specified ([conc. +/- [error] units). Here the column integrated concentration specified over the area of view is specified in number/area while the error is specified as RMS residual. The information on concentration and errors may be specified in Fig. 14 as well.

6.     It is normal practise to show data and fit of the target pollutant spectrum as the proof of detection and quantification. Here, in Figure 14 the data and fit are shown, however, if contribution from all other gases and baseline offset is removed, the fit for NO2 retrieval will be clearer (compare with Fig. 13 in General et al 2014).

7.     Equations (2) and (3) in section 4 clearly depend on the viewing angle. If the angle of view is high (for SCD), the VCD retrieved will be erroneous because it may overestimate. Is there is a limit of azimuth angle where the measurements might be deemed valid beyond which the correction may be deemed erroneous? If so, the limit may be specified.

8.     The IDOAS measurements are for the 1 km column integrated values of NO2 whereas TROPOMI gives full atmospheric column integrated concentration. Do we expect both to match? Would the measurements from a different height give different results?

Reviewer 2 Report

Comments and Suggestions for Authors

DOAS system has been widely applied to monitor air pollution.This manuscript introduces an interesting airborne fiber imaging DOAS system and the retrieval algorithm for tropospheric NO2 column.  The optical signal acquisition module of the system combines a telescope and an optical fiber transmission unit, which separates the telescope and the spectrometer, so the IDOAS has the advantage of easy application in aircraft. The comparison between the airborne measurements above Tangshan City and the satellite data suggests that the airborne fiber IDOAS has the capability of detecting tropospheric NO2 accurately. The structure of this manuscript is well organized.  Minor revision is required before publication.

Specific comments:

1More details are needed on how to make the coupling between telescope-fiber-spectrometer after the optical system used the optical fiber transmission unit in Section 2.

2Please describe how to ensure that there is no crosstalk effect between the multiple fibers of the optical fiber transmission unit In Section 2.

3Due to introduction of fiber optics, the CCD intensity distribution map showed bright and dark stripes. Please describe how to reduce the influence of dark stripes in the Section of data processing.

4In Figure 18, points 4 and 5 are not clearly labeled, please relabel them.

Comments on the Quality of English Language

Minor revision of English writen is requied.

Reviewer 3 Report

Comments and Suggestions for Authors

The paper deals with the exciting topic of airborne measurements over the city of Tangshan in China. 

Only a few comments: 

  • It would be nice to use the color scale in Fig 6 – 9 regarding the aircraft height.
  • Please comment, what are the errors due to flight height, if measurements were done on 500m higher or lower level, can we expect different VCD (as shown in Fig 22 a
  • What is the desired flight height for taking the most accurate VCD? Or reasonable to make a comparison with Troponomi?
  • How are wind speed and aircraft speed influencing the results, is it possible that different vertical wind profiles and vertical wind speed can have an influence on VCD, not only the cloudiness?
  • How are the authors explaining discrepancies in the final results of VCD in Fig 22a, and how can those results be improved (e.g. increase correlation coefficient)?
